# Improved Interaction of BIM Models for Historic Buildings with a Game Engine Platform

Yu-Pin Ma

Department of Architecture, National University of Kaohsiung, Kaohsiung 81148, Taiwan;
yupinma@go.nuk.edu.tw

**Abstract:** For the purpose of the conservation and representation of cultural heritage, the development of a visual interactive environment to provide an integrated application suitable for collecting a wide range of big data information from historic blocks is regarded as an effective solution. In this study, the existing modeling of wooden building information is imported into a game engine as a practical study for the design of a user-friendly interactive interface design, a Unity integration platform is built, a Revit external application is developed, and cloud services architecture is set to solve the transmission and integration of information models between different platforms. In addition to strengthening the future application of historic building information modeling, this visual information-based interactive environment can help enhance communication and subsequent management of repairs, and bring opportunities for sustainable management and diversified applications to the research field of historic building restoration.

**Keywords:** historic building conservation and representation; game engine; building information modeling (BIM); information visualization; Human-Data Interaction (HDI); user interface design

## 1. Research Background

How to apply digital technology to preserve the background and footprints of historic buildings during their conservation is an important topic, and new opportunities appear with the rise of the emerging digital technology. The establishment of a semantic BIM model for historic buildings and parameterization of its component information are regarded as effective solutions to enhance future applications of BIM models for historic buildings, and they are an opportunity to initiate sustainable management and diverse applications.

In addition to providing accurate recording and realistic presentation, 3D visual information-based modeling is an important tool for analysis, reconstruction, and virtual presentation, and can be used to expand the user's ability to explore and interact with the historic block's environment. Regarding the conservation and representation of cultural heritage, in addition to simplifying integration and facilitating a better understanding of heterogeneous information, the visual interactive environment is developed to provide integrated applications with a wide range of information and large amounts of data suitable for historic blocks. However, in spite of the sufficient information on the details of urban blocks and the volume of buildings, BIM research is mostly focused on the study of 3D model visualization; thus, users lack the capability of interacting with and feeding back the surrounding environment in real-time.

In recent years, virtual reality technology has been expanded to apply to cultural heritage conservation and representation, which provides users with an environment to interact with and give feedback regarding scene objects in real-time, and thus, presenting new possibilities for the workflow and cognitive mode of cultural heritage conservation and representation. Through its real-time visual control and virtual reality performance, VR has been proven to provide an effective communication platform [1] and many scholars are currently conducting research on importing established BIM models into an environment that users can navigate and interact with [2].

Among them, the application of game engines in 3D model visualization has gradually become a research trend of concern [3–8], as it provides a high degree of interactive experience, as well as effective communication and an integration environment. Moreover, as the BIM model contains a lot of geometric and non-geometric information, it is the perfect candidate to be imported into the game engine to build a realistic virtual environment.

As an effective solution, this study selected an open-source game technology platform to import the semantic BIM information model into the game engine platform, which focuses on how to import the existing BIM model for historic buildings into the engine platform, optimizes the restoration process of historic buildings in conjunction with the development of an interactive program, and intensifies the operation and presentation of historic building environments with the aid of game technology. In addition to expansions and making access to data more convenient, game technology can be used for intuitive interaction and management of historic scene information to make best of restoration information, and future applications of historic building models can be expanded during the restoration design, which will reduce the probability of misunderstandings during communication.

## 2. Literature Review

The conservation of historic buildings and cultural heritage is a long and unique process, as it requires innovative methods and technologies to promote restoration and management. Among the processes, the integration and representation of information of historic buildings have a unique factor that does not exist in other fields, that is, the multi-layered spatial character of the third or fourth dimensions. Therefore, in an attempt to maintain the original appearance of the historic building environment [9], during the relevant literature collection and research, participation in and communication with persons in different majors are required, which is supplemented by visual building information modeling (BIM) with Semantic. This method can also help to understand the as-found state and specific features of historic building environments through different spatial information specialties and modeling platforms, as a large amount of qualitative and quantitative heterogeneous information is generated [10], including 2D and 3D geometric models and non-geometric information. While it is very important, it is also time-consuming to summarize and represent all the heterogeneous information collected and generated.

In fact, it is not an easy task to build a general 3D visual building information modeling (BIM) platform for the effective management of heterogeneous information in historic building environments that can integrate rich semantics in 3D models and represent an interactive interface scene that is both intuitive and user-friendly [11]. According to the abovementioned research issues, the important research goal of this study is to build a 3D interactive information management platform that can access historic scenes and build information models applicable to various scales that will allow both information queries and virtual tours.

With the rise of virtual reality/augmented reality (VR/AR) platforms and game technology, the application of BIM offers more possibilities. To assist communication and use, some scholars have proposed the concept of integrating the BIM model with virtual immersion technology [12], which will facilitate further development of the virtual environment, and the real-time rendering and interactivity capabilities that BIM lacks can be expanded to allow design expert teams to more easily understand issues during discussions and meetings. Meanwhile, the Unity3D game engine also provides a more user-friendly development environment and exhibition possibilities for the application of BIM to actually simulate the interactive environment and optimize the application and discussion of urban and architectural design research [13].

### 2.1. Parameter-Based and Visual Application of Historic Building Information

The 3D visualization of building information modeling (BIM) has become a key factor in communication [14]. Throughout the process of building production and construction, and the visual modeling of the same information rich model, all project participants,

including experts and non-experts, can discuss the design in a common 3D model, and such interaction can enhance collaboration and understanding among participants and improve the old workflow. BIM technology has currently been extended to the restoration and conservation of historic buildings and other related issues, and many studies have used BIM technology for the investigation of historic buildings, for the early stage of modeling and analysis, and for management and simulation reproduction after modeling.

Unlike new buildings, historic buildings are existing buildings that must be reconstructed with reverse engineering technology; thus, some research has digitalized historic buildings to collect space information by modeling with point cloud information, as obtained from 3D scanning in combination with 3D information modeling [15]. Such models are transferred into structured and parameterized visual models, which can provide useful information to experts including designers, architects, owners, and historians, and are very useful for traditional building construction management.

Architecture and urban design have always focused on the concept of visualization and materialization. Visualization through building information modeling (BIM) can interact with models in a fast and intuitive manner. Although BIM-related applications have been widely used, there are still challenges in the costs and time resources required to develop and maintain the BIM models of historic buildings. Moreover, user proficiency in BIM and data interactivity are also obstacles in the development. In order to overcome these challenges, the first issue to address is data transmission. It is pointed out that data transmission may be solved through parameter-based external tools (for example, Dynamo) or Application Programming Interfaces (API), as they may act as information exchange links between BIM models and external data sources. In addition, the processes of the automatic maintenance of the model and data transmission may also reduce the costs for model-related settings and maintenance [16].

Boeykens et al. [17] proposed a virtual environment restoration system for the reconstruction and restoration of synagogues, which emphasizes the use of parameter-based and adaptive controls to effectively reduce the workload of future reconstruction and to provide a new technology and opportunities for the conservation of historic buildings.

Ma, Y.P. et al. [18–20] proposed a system incorporating BIM to aid the traditional restoration process of historic buildings. The system records the restoration process information in the building life cycle and builds a BIM model containing building health records and the restoration information of components to assist in the traditional construction process of traditional historic buildings, as well as the communication process between construction end and the design end. In addition, Ma, Y.P. et al. developed plug-in applications that can instantly return information to the construction end. Meanwhile, the visual models and parameter-based component design improve the efficiency and smoothness of the repair design process and construction communication process, which increases the continuity of the life cycle of historic buildings and the possibility of preserving cultural heritage.

In addition, England has acquired a set of knowledge and experience in the application of BIM to historic buildings and put forward a series of practical guidance items for the detection, management, and care of historic buildings and their environment, as well as professional technology regarding the surveys, measurements, and recording of historic buildings. This proves the potential advantages of the application of BIM on historic buildings, including assessment of conflict detection, quantitative cost estimation, building simulation, project management, facilities, and asset management, and shows great potential for the management and conservation of historic buildings [21].

While there have been numerous papers and applications presented that introduce the conservation and restoration of historic buildings into BIM technology, there are still deficiencies in its application to restoration works. The lack of an intuitive visual interactive environment to facilitate discussions and communications between the construction professionals and restoration designers results in a failure to meet the communication demands. Moreover, different software tools operated by users may lead to various problems, such as information errors, the lack of an effective communication platform, and obstacles

in interaction and cooperation, which have high time consumption, and thus, affect the restoration and conservation process of historic buildings.

According to the above research, this study found that there is still room and potential for development in terms of importing BIM technology into a historic building model, which would increase the connections and interactions between users and information, and promote the application of visual repair information. However, for the purpose of expanding the applications and interactive function development of the BIM model, a development environment suitable for applications and simple method of information transmission are required, as the BIM software commonly used in the industry does not provide a development environment, and the format and structure of a BIM model are different from the development platform commonly used in the industry.

### 2.2. Combination of Building Information Model and Virtual Reality Technology

In recent years, building information modeling (BIM) has been expanded to virtual reality (VR), which is an immersive environment generated from computers that can be adjusted and manipulated instantly by users. Different from the traditional BIM platform environment, VR can interact with various architectural components and provide users with a new visual space experience and project communication mode; thus, it is considered the most promising approach to the workflow quality of the overall Architecture and Engineering Construction (AEC) and Facility Management (FM) [22]. Moreover, real-time visual technology and virtual reality technology have been proven to provide an effective communication platform [1], and many experts and scholars have begun to study the integration of information model technology and virtual reality technology.

Johansson et al. [12] used Autodesk Revit software to develop a plug-in capable of real-time rendering, which integrates the BIM model with virtual immersion technology through a head-mounted display, and allows design discussions at any time and place. In addition to providing detailed information about objects in the environment, it also enhances the rendering performance, navigation interface, and real-time interaction functions that BIM software lacks. Through the navigation interface, regardless of a lack of relevant background or professional knowledge, users can freely browse 3D scenes from the first-person perspective, and thus, even inexperienced users can easily interact with the model in a fast and intuitive manner when discussing design.

Kieferle and Woessner [23] proposed importing the BIM model into a collaborative visualization and simulation environment (COVISE). By integrating simulation, manipulation, and visualizations in a seamless manner, users could easily learn how to control the system, browse the virtual reality cave (VR-CAVE), and have a greater sense of presence when directly connecting the actual walking experience during browsing. With the above-mentioned increased sense of presence, users could be more integrated into the immersive environment, and the content of the space could be more clearly presented.

While BIM software is usually run independently, the virtual environment and inter-activity capabilities that it lacks can be made up by combining it with the virtual reality technology; however, the model and underlying database must be updated constantly. In most VR environments, design changes cannot be displayed in a timely manner, as they lack an automatic and efficient data transmission method between the model and the data. By considering the time consumption and the complexity of the VR environment during the integration of simulation, reproduction, and visualization, Du, J. et al. [22] developed a VR display information design that can instantly and automatically update information. In this design, the data are transmitted through cloud data based on a cross-platform cloud, where BIM data are synchronized in real-time in VR environments, and an innovative data protocol is entered.

However, as a large number of resources and different professional backgrounds were required, the above relevant research indicated that the integration methods of virtual reality and BIM were still limited in many aspects. In order to solve the abovementioned integration problem, numerous relevant researches have been initiated to discuss the large number of

information models contained in the BIM model, which could be imported into a game engine as a third-party platform to build a realistic virtual environment to achieve the best integration, and reduce the difficulties and restrictions encountered during development. In the following, this study further explores data integration for the Unity 3D game engine, as well as the cases related to developing the interactive environment functions.

### 2.3. Application of Game Technology to Optimize Historic Building

As the digital preservation of historic buildings has become a trend with steady development, historic building and old city block scenes have evolved into new display applications through the introduction of game technology, which combines virtual augmented reality and game technology to improve users' spatial experience of historic buildings.

Boeykens et al. [3] combined Unity and ArchiCAD to reconstruct historic buildings and simulate scenes by combining the game engine and BIM model. The information models of historic buildings can also be further edited and re-imported to the game-based platform, where all materials can be adjusted and interactive applications can be fully presented. The visual presentation of a historic building is more realistic through the game environment, and heterogeneous data are more accepted, which makes it easier to develop and apply.

Albourae [24] developed an integrated application platform for historic buildings that integrates BIM, GIS (Geographic Information System), AVR (Augmented and Virtual Reality), and game technology. First, the historic building is thoroughly surveyed and researched, and the survey and research results are integrated with the geographic space information system to build a 3D information model with high details and location information for historic buildings. Through the development of the game-based platform, users are provided with a rich visual model with interactive features for a more immersive experience supported by AVR interactive virtual technology. Upon the integration of BIM, GIS, AVR, and game technology, in addition to being operated in different locations, the application can be easily developed and applied.

As BIM software is mainly used to design new building modeling, the threshold for modeling historic buildings and complicated components is high, and it is difficult to present the original appearance and delicacy of traditional components. Therefore, Ma, Y.P. et al. [4,5] selected SketchUp software, as it has a low threshold, to build a historic building model with high details, followed by building the attribute data of the historic building volume with GIS. Then, the SketchUp building model of the historic block, as well as its corresponding GIS attribute data, were imported together into the Unity 3D environment to integrate the model information and present the virtual scenes to preliminary complete information integration and complete the function development of the interactive platform. Users can click the mouse to select the required visual information in the platform scene, and the data regarding the historic building and old city blocks can be updated (unilaterally) by uploading the data to the platform through the GIS attribute table.

The above research shows that, in addition to strengthening the visibility of model information, the game-based platform environment is also relatively friendly for importing and integrating heterogeneous information from different platforms and it is highly developable for user's interaction experience in the historic building model, while the operating interface is user-friendly. Whether users have professional or non-professional backgrounds, they can intuitively operate the platform for restoration surveys, environmental navigation, and historic architectural space experiences.

Regarding the advantages and diversified use of importing game technology into historic building restoration, the above case study found that many scholars choose to use the Unity game engine as the integration tool, and argued that the use of the Unity game engine for expansion of historic building modelling is highly feasible, and can bring about high-quality visibility and flexible manipulation. Through the simulation of real scenes, the user's awareness and understanding of the space are intensified; moreover, the

game engine accepts different data and conversion formats from different platforms, which makes development easier and with fewer restrictions.

Moreover, our research team has exhausted a lot of experience due to two related previous works [5,20], mentioned in the literature section of this article. The former [5] completed the development of the Revit plug-in interface API, and the latter [20] tried to import the information model into the Unity platform for the integration of geometric models and non-geometric semantics. The purpose of this research is to continue the accumulation of related research. This research uses the former's Revit information model [5], based on the former's experience in developing the Revit API [5] and the latter's experience in importing the model into the Unity platform [20], and tries to combine the Revit API with the Unity platform and develop an interactive interface.

### 3. Research Topic and Purposes

#### 3.1. Research Questions and Topics

First, regarding the topic of this study, historic buildings are important to cultural heritage, and there are many difficulties in their restoration design process, including constant communication and discussion; thus, mutual cooperation is required between the design side and construction side to avoid errors in the restoration process that may affect the cultural significance of historic buildings. However, as the traditional workflow is time consuming and inefficient, and can easily result in errors during information exchange, the conveyance of traditional 2D drawings and 3D models will prolong the workflow. Therefore, building a semantically rich BIM model is regarded as an effective solution for restoring historic buildings and parameterizing their component information.

However, the BIM model of historic buildings is still insufficient for practical application at the restoration site of historic buildings. More than 60% of those on the list of historic sites and historic buildings in Taiwan are traditional buildings with wooden frames as their main structure system; thus, traditional historic buildings with large wooden frames take the column and beam system as the main structural system. As buildings with wooden frames are not easily understood, and due to their large number of components and complex organization, they are not easily judged, and the BIM model lacks a fast and intuitive way of reading information.

Moreover, regarding the restoration of historic buildings with wooden frames, after restoration, the architecture should be dismantled for damage investigation before reorganization. Since the BIM model lacks intuitive visual presentation and an easy-to-use interactive communication environment, difficulties often emerge in communication between the design end and construction professionals, which hinders restoration efforts. Therefore, as the BIM model for historic buildings still lacks an environment that can interact with the model, new possibilities can only be provided to BIM by importing them into the development of other environments.

#### 3.2. Study Subject and Objective

In summary, this study imported the BIM model of semantically rich buildings with wooden frames into a game engine for development and research of its interactive environment and takes historic scenes in Tainan as the research field, which aims to solve the problem of the insufficient interaction and communication of the historical building model, assist participants in communication and cooperation, ensure the accuracy of information during restoration, enhance management ability and restoration quality, reduce errors and communication difficulties through the design of an intuitive operational interface, and further develop the interactive function of the application.

This study used the BIM model for historic buildings with wooden frames, which was built and completed by our team during the initial study of Huji Palace in Tainan [10–12], and relevant restoration and case diagnosis information were added into the model components. Parameters provided include the deterioration of component parts, the restoration method, restoration time, intervention level, restorers, and other non-geometric information.

This study focused on how to expand the development and application of interactive ability within the parameters of existing BIM models of historic buildings. While this study does not discuss the modeling process, the detailed contents, or the back-end repair construction method, it places emphasis on the development of interactive scripts using existing information models to enhance the ability of information amplification and interaction.

This study aims to develop a visual interactive operation and control environment that can promote the management capability for historic buildings and cultural heritage sites by using game technology to assist the integration of heterogeneous information in a historic scene, conduct practical research of the user interactive interface design, and enhance the visibility and interactivity of the BIM model of historic buildings. In addition to assisting communication and ensuring the integrity of transferred information, this visual interactive operation and control environment can also be used to optimize restoration at the construction site and to strengthen the application of historic building information modeling.

The tasks expected to be completed under this study are as follows:

1. Simplify visual operation: The user is provided with the functions of simple navigation, information query, or integration of different model information in the environment; the user can read the component restoration information by clicking and removing any inconsistent judgments, as caused by the complexity of the component information. Visibility and future applications may be enhanced through an intuitive presentation of visual information.

2. Enhance interactive communication: Animation is provided for the disassembly and composition of historic building components. Animation of disassembly with easily understood operation orders is made based on the consensus record of the architect and the survey staff of the research unit to assist participants in communication and use at the restoration site.

3. Overlay automatic information bi-directionally: the bi-directional information transmission function is provided through the plug-in application interface design, to ensure that model information can be accurately transmitted between building information modeling software (Revit) and the game integration platform (Unity). The restoration information of historic buildings is accumulated in the restoration record over time, which makes the integration process more logical and easier to operate, and the restoration information of different components during different periods can be read through selection by clicking.

### 3.3. Research Method and Procedures

By importing game technology, this study aims to enhance the interactivity of the BIM of historic buildings, develop the integration platform, and strengthen the application of the BIM model for historic buildings. Enhancing the link and interactivity between the restoration participants of historic buildings and historic building information can solve the problems of the insufficient interactivity and communication abilities of the BIM models of historic buildings. This study also proposes the standard workflow for the application of the BIM models of historic buildings to the game engine, which can make up for the shortcomings of the application of the BIM models at the restoration site, and extend the possibility of more diversified applications of the BIM models of historic buildings.

The game engine Unity, which has a high ease of use and development freedom, is selected in this study as the platform for information model integration and interactive environment development due to its advantages of quick import of the geometry model and the integration of the geometric information. With the relatively simple development and relatively friendly interface application design, the Unity platform can provide a highly visual simulation environment to enhance the user's experience of the space. Moreover, Unity can provide more value in information use and lower the user threshold to interact with information.

In order to achieve the above objectives, we conducted program development and platform construction in Section 3 of this study and proposed the method of information

collection and programming integration. We also simulated the use scenario of the Unity integration platform, determined the purpose of using the platform, and wrote interactive scripts to complete the platform pre-construction. In the following Section 4, this study conducted an interactive interface design, set up the visual and interactive information environment in the Unity integration platform, and simultaneously developed and designed the plug-in programming interface in Revit to facilitate accurate information transmission between Revit and the Unity integration platform, in order to make the integration process more logical and easier to navigate (Figure 1).

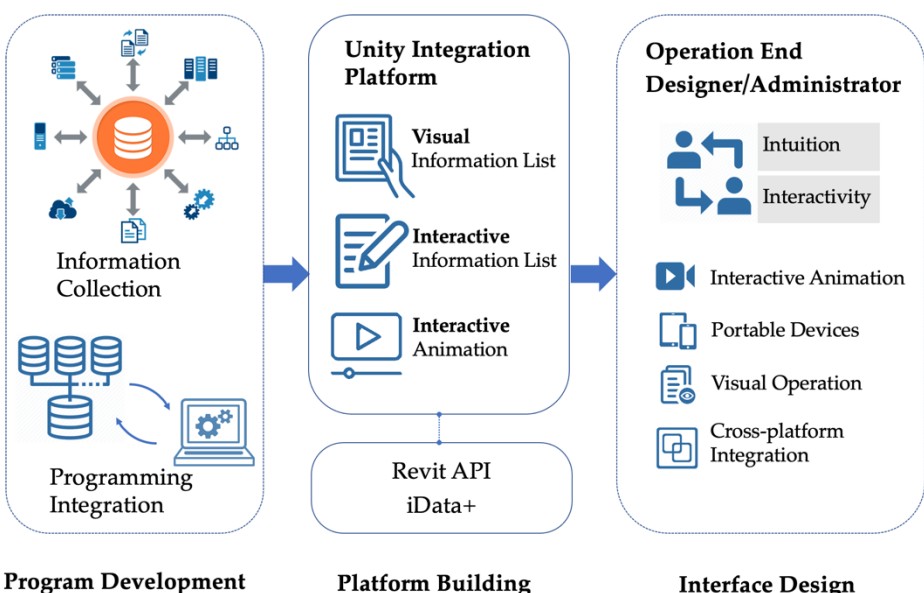

**Figure 1.** Flowchart of interactive platform construction and interface design.

## 4. Program Development and Platform Building

This section is divided into two stages of development: Information Collection and Programming Integration (Figure 2). According to the preliminary findings, if the BIM models of historic buildings are directly imported into the game engine, the model should undergo format conversion before being imported into the game engine, which is prone to information omission, meaning it would lose the information-rich advantages of the BIM model and the significance of platform development. Therefore, this study proposed an information collection method to accurately extract the restoration information from the BIM model of existing historic buildings and added new information fields required for classification and sorting, to ensure that the non-geometric semantic information in the BIM model can be retained as completely as possible during conversion for import into the Unity platform. Then, the extracted geometric model and non-geometric semantic information in the Unity3D game engine environment were re-integrated at the following programming integration stage to improve the accuracy of information conversion across platforms.

The information collection work is divided into two projects (Figure 2): collection of the geometric model (refer to the circle number 1 marked on the left in Figure 2), which involves the export of material and volume, and the collection of non-geometric semantic information (refer to the circle number 2 marked on the left in Figure 2) in geometric models, such as restoration records, model data, and other information, which are both exported in effective formats and imported into the Unity 3D game engine together for integration (refer to the circle number 1 and 2, marked on the upper right in Figure 2). Information collection can reduce the omission and unsupported formats of information caused by cross-platform conversion during the import of information models into the game environment.

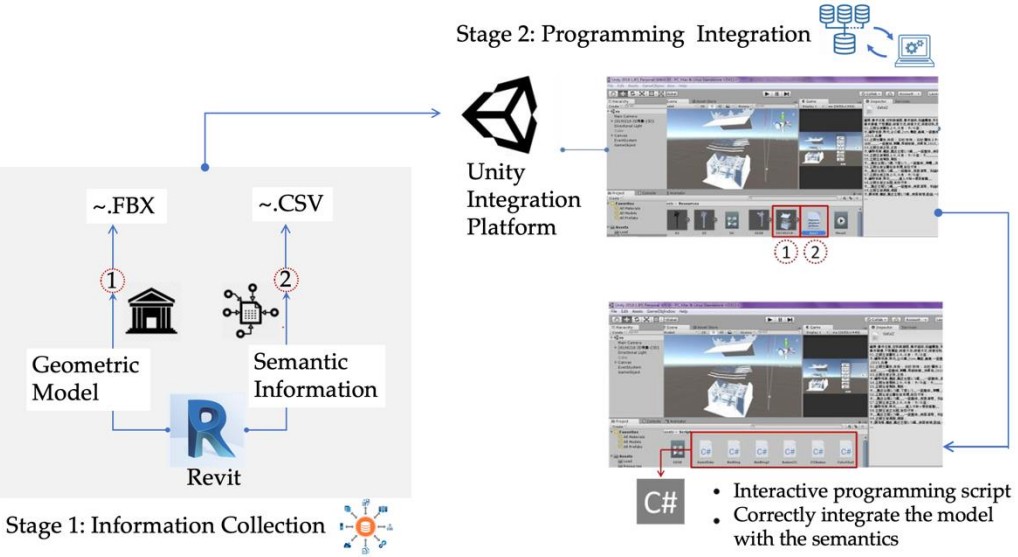

**Figure 2.** Flowchart of program development and platform setup.

This study employed Dynamo, the plug-in software of Revit, to collect non-geometric semantic information at the experimental stage, in order to determine the standardized conversion principles, such as the file format and unit required for importing BIM into the game engine through preliminary experiments. The Dynamo plug-in has the advantage of directly capturing the parameters of Revit model components and writing the behavioral scripts in a visual manner. Moreover, the scripts can also be written in a literal programming language, which is easier to use and serves as references for information structure and formatting for further development of advanced information collection solutions.

In addition to information collection, Dynamo can be used to import the information to be updated in the Unity platform into the Revit model again in the Excel format (Figure 3), and the information can be updated in both directions. However, for both information collection and subsequent updating, the Dynamo plug-in program must be re-executed for each collection task, and no expected action will be completed until the execution button is pressed, which is relatively inconvenient for the operator, and renders the workflow more complex and the subsequent application development function less free.

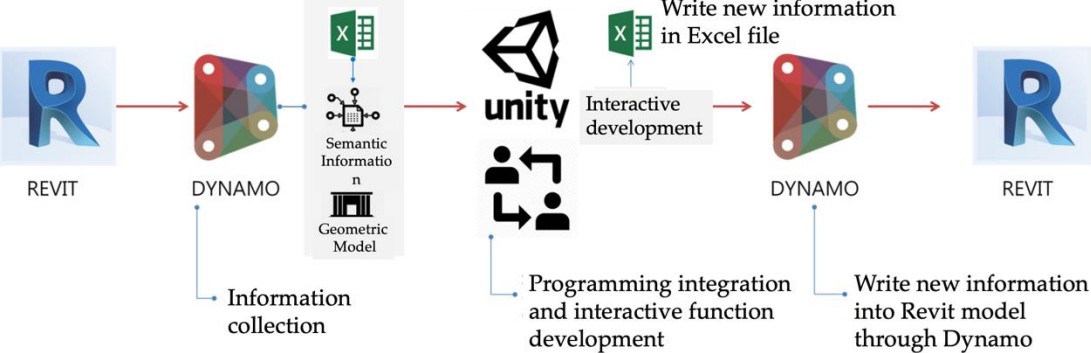

**Figure 3.** Flowchart of Dynamo's information collection and bi-directional updating.

Therefore, Visual Studio was used for advanced development at the second stage of this study, where the information collection package of non-geometric semantics is subject to parameter collection and format conversion by capturing the parameters of the components through the Visual Studio program of Revit's plug-in programming interface (API) (Figure 4). Thus, users only need to click the export model information button built in the Revit's plug-in application panel to complete information collection, which simultaneously uploads the semantic model information to the cloud service framework

(WebAPI) (Figure 5). This function simplifies the operation and workflow, solves difficulties in application and reduces the operation threshold, simplifies the information collection process, makes it easier for the user to operate, makes information transmission more direct, preliminarily builds the follow-up interactive functions, and strengthens the application of the front end.

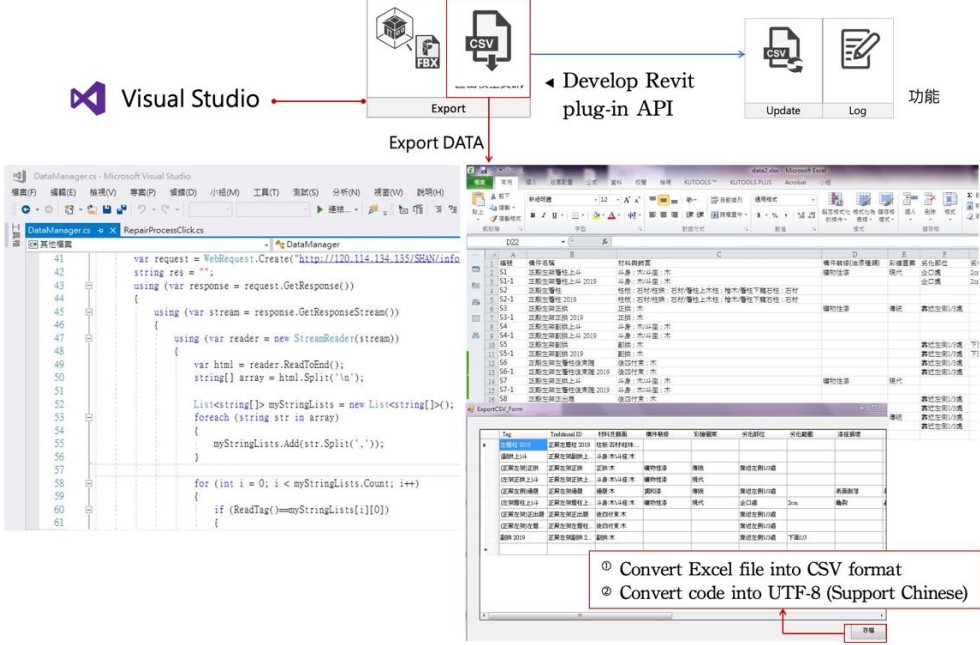

**Figure 4.** Advanced information acquisition process of Visual Studio.

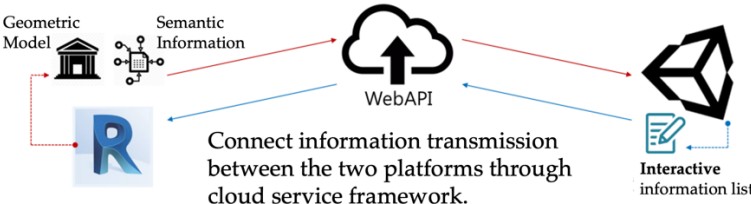

**Figure 5.** Cloud service framework (WebAPI) of Visual Studio.

WebAPI helps in enabling the development of HTTP services to reach out to client entities such as browsers, devices, or tablets. The function of Web API here can not only reduce the burden of computer calculations, but also instantly update the model information to the integrated platform. The following figure (Figure 5) shows the transmission process between the Revit model information and the Unity integrated platform. The model information collected during the collection phase can be uploaded to the cloud storage space in real time, and the model information in the cloud can be automatically captured during the programming integration phase. In addition to strengthening the connection between the information model in Revit and Unity integrated platform, the cloud service also increases the applicability of model information.

Completion of the information collection at Stage I was followed by the programming integration stage (Figure 2). This study used the Unity 3D game engine as the integration platform to build an interactive environment for the information models of historic buildings, where the geometric model (.FBX) (refer to the circle number 1 marked on the left in Figure 6) and non-geometric semantic information (.CSV) (refer to the circle number 2 marked on the left in Figure 6) collected in Stage I were imported into the game engine, and the geometric model and corresponding non-geometric semantic information were correctly integrated through Unity C# programming language (Figure 6).

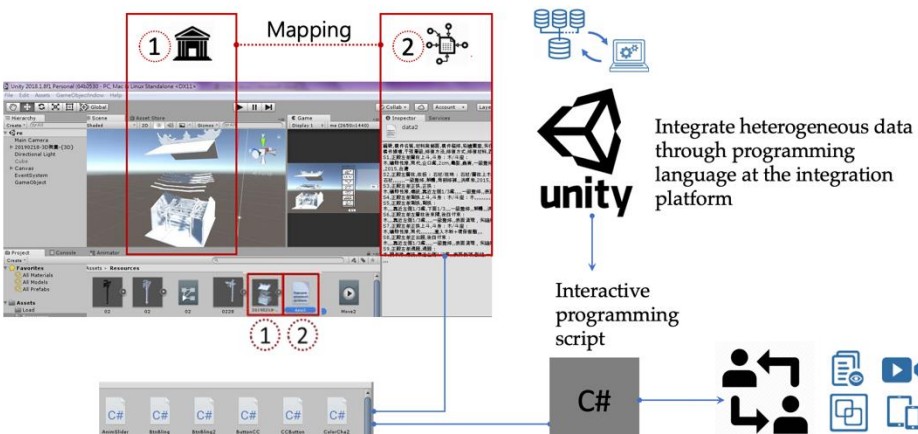

**Figure 6.** Flowchart of Unity programming integration.

After the above stage of information collection and programming integration, the BIM models of historic buildings were imported to the game engine, and a link was built for the geometric model and the non-geometric information. Then, the restoration information of the historic building was overlain, updated, and edited to strengthen the application capability and back-end management, which made the restoration of historic buildings more visible and superseded the time-consuming exchange of traditional paper information, and such continued information conveyance constantly expanded back-end management.

The integration platform proposed in this study is a customized project, which is developed using the existing BIM model [18–20] that has been completed before. Since the original model was built using the Revit 2017 version, if it is higher or lower than this version, the program and platform developed by this research will not be able to be used. It is also difficult to have commonality in different historical building components, which restricts the versatility of the integrated platform. At the same time, this study also found that the file capacity of the model imported into the game engine platform has a limit of 1 GB, so the model cannot exceed this limit.

In addition, the coordinates in the Revit environment are the right-hand rule, which is also the most common coordinate system with the Z-axis facing up. However, in the Unity3D game engine, it is based on the left-hand rule with the Y-axis up coordinate system. The coordinates between the two platforms are different. After the geometric model is imported into the game engine, the relative position information between the model components is different from the original position, so that the model cannot be reorganized and presented in a correct way. Therefore, it is necessary to convert the coordinates collected from the Revit model components into the same coordinate system as Unity. That is, the Y-axis field and the Z-axis field are interchanged, so that when the Unity environment is subsequently entered for programming integration, the position information and the geometric model can be correctly integrated and reorganized and presented in the correct position

## 5. Interactive Function Development and Plug-In Application Interface Design

After the front-end program development and platform construction were completed, the interactive function development and user interface design included two parts: the interactive function development of the Unity integration platform and the design of the Revit plug-in application interface (iData+), which had the objective of making the system user intuitive and easy to operate, meaning it increased the interactive features lacking in the original information model, including the ability to write an interactive script, simulate usage scenarios, design the interactive interface, animate the disassembly and assembly of components and other items, and complete the function and application of the integration platform, to ensure that users can interact with the information model more intuitively. In addition to assisting in communication during restoration design, it also strengthens the application and value of the model of historic buildings and restoration information.

### 5.1. Interactive Function Development for Unity platform

For more adequate applications of model information, the Unity platform was selected in this study to further develop the interactive function and the design of the user interface. By focusing on enhancing the link between the user and the information, development was divided into three functional projects for functional development and interface design, namely the "visual information list", "interactive information list", and "interactive animation".

Taking users' implementation of the integration platform project at the restoration site as an example (Figure 7): (1) The user can review the information of any restored components or components awaiting restoration in the project through the "visual information list" option (refer to the circle number 1 marked in Figure 7), conduct comparisons of the actual components at the restoration site, and browse components at various stages of restoration. (2) When the user finds any error in the component information and has to revise the component information or add restoration suggestions, the user can input and store the updated information content through the text box in the "interactive information list" (refer to the circle number 2 marked in Figure 7), and the updated information will be automatically uploaded in real-time to the cloud service framework (WebAPI). The restoration information may be added or overlain continuously through such lists and may be compared with the restoration information of the same component at different stages in the list. Upon completion of information updating on the platform, the user updates the Revit model information through the plug-in programming interface (iData+) of the platform administrator. (3) Since the Revit platform only has the function of viewing the model, this platform provides the application of the "interactive animation" option (refer to the circle number 3 marked in Figure 7). In addition to providing simple animation of the camera path, it also provides interactive animation of the disassembly and assembly of building components, which is more in line with the needs of restoration designs for historic buildings and assists in the communication between the design end and the construction end. In the future, its function development may be customized according to the user's needs.

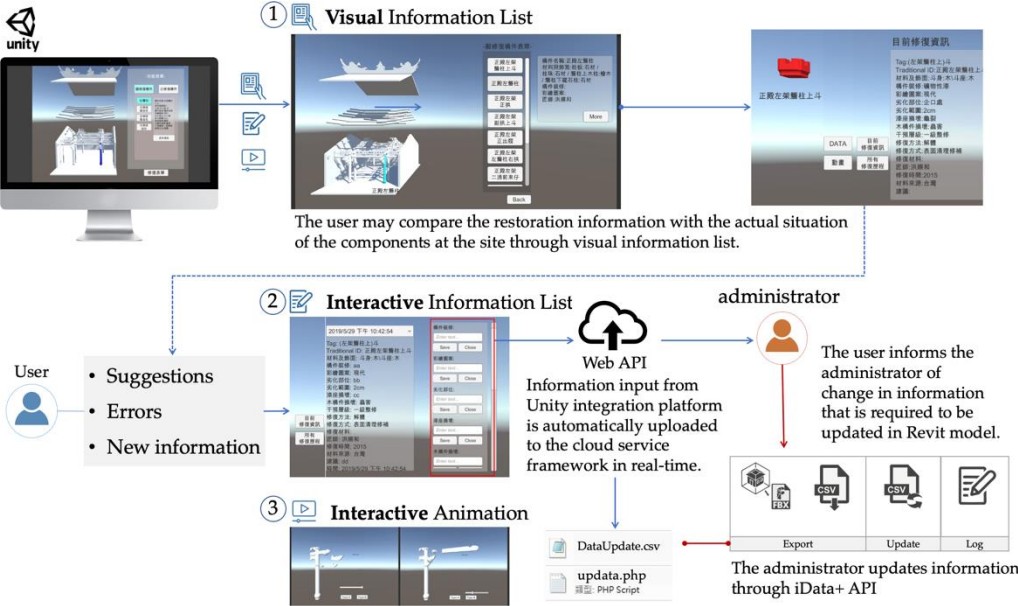

**Figure 7.** Operation work flow of the integration platform by the user at the restoration site.

### 5.2. Revit's Plug-In iData+ Application Interface Design (API)

This study developed a plug-in application interface (API) named "iData+" in Revit software and imported the Visual Studio from the information model into the Unity integration platform, which was subsequently returned to the Revit software for information updating, and thus, reducing the difficulties and barriers in the platform operation and software, and simplifying the workflow to the maximum. Currently, three function panels

are set on iData+ API: "Export model (Export)" (refer to the circle number 1 marked in Figure 8), "Update model information (Update)" (refer to the circle number 3 marked in Figure 8), and "Revise model Log (Log)" (refer to the circle number 4 marked in Figure 8). This panel contains four function buttons, and new functions may be added to the platform management end in the future. The functions and applications of the buttons are explained in the following paragraphs (Figure 8).

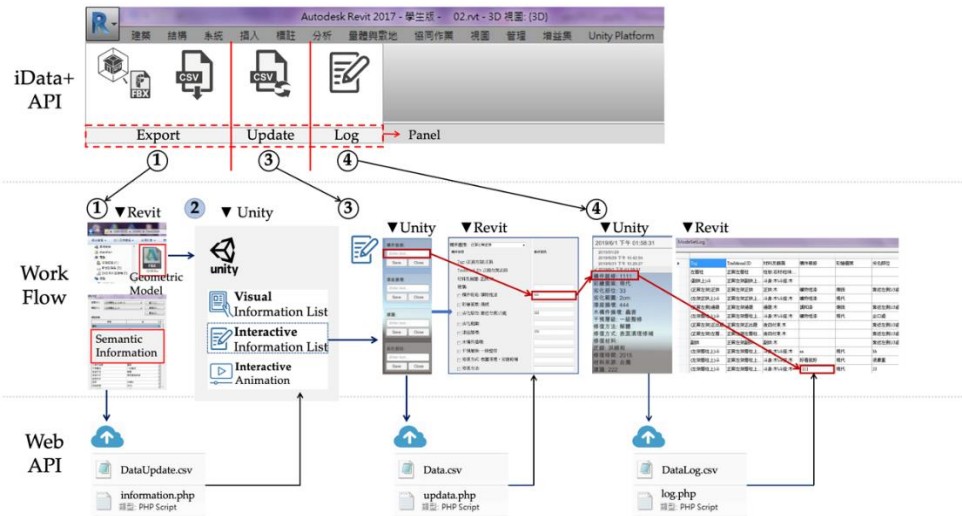

**Figure 8.** Use flow of IData + application interface.

The function of the iData+ interface replaces the original manual work, meaning users no longer need to open the Dynamo plug-in software for information collection, for changes to the model, or for updating information, meaning they can export the model information and semantic information more easily and quickly by pressing the function button in the operation flow of the iData+ interface panel. In addition to developing the function key to export the Revit model and semantic information, the function to update model information is also built. When the interactive information list in the Unity integration platform is directed back to the Revit model, the information is updated bilaterally and the restoration information is accumulated as the restoration log; thus, the information of historic building models is enhanced.

When the historic building model is changed or new components are added, the model and semantics can be uploaded to the cloud service framework and Unity integration platform through the "Export model" and "Export Model Information" buttons, which are located in the first Export function panel. The buttons have the following functions: after pressing "Export Model Information", the prompt box and tips text will pop up to export the model into the specified Resource files in the FBX format; after pressing "Export Model Information", the program will automatically capture all the parameter information of components in the Revit model, and the information box will pop up and display in the exported information panel; after pressing "archive", the collected model information will be uploaded to the cloud service framework and stored in the CSV format, which supports display in Chinese characters. In the case of an updated geometric model, the administrator only needs to pull the geometric model into the game environment, after which the program will automatically grab the restoration information corresponding to model components, which will be stored as the specified Unity file for use at the restoration site after completion of the programming integration.

When the user adds or changes component information through the Unity interactive integration platform and informs the platform administrator to update the Revit information model, the platform administrator can execute the "external import" function through the "Modify Model Information" button in the second Update panel, which loads the updated Revit model to the cloud service framework, and the display panel of new

components will pop up. The administrator will click the updated information to confirm the information to be written in the panel, and then press the "Confirm" button, after which the revised content will be automatically uploaded to the cloud service framework. The relevant records can be extracted through the "Model Revise Record" button.

As mentioned above, all revised information is recorded to the list according to the time, and logged by the "Model Revise Record" button in the last Log function panel. The platform administrator can export the overlain information and restoration information log in the Excel format of the revised model record for more value-added applications through word processing software. The Archive function is also set, to ensure that the restoration record can be stored at the local side, meaning it can access more applications, including editing in the Revit software, which renders the information more valuable.

As shown in Figure 8, the function of assisting two-way updating through the WebAPI cloud servive and iData plug-in is divided into three parts. The first part is to automatically store the model repair information data collected during the information collection phase in the cloud service storage space. The second part is to upload the new information entered by the user in the Unity integration platform, so that Revit can grab the updated information. The last part is to record all the modification history in the platform, all updated information history can be displayed through the form in the Revit software.

### 5.3. Platform Usage Evaluation

The purpose of this study is to improve the interaction of historic building models and the application of information, design the standard workflow of importing the BIM models of historic buildings into the game engine, build the Unity integration platform and Revit's plug-in application interface (iData+), and connect those two platforms through the cloud service framework. The information can be automatically transmitted in both directions and can also be overlain. In addition to reducing the threshold of software operation, this function provides a simplified process for the back-end platform administrator. The user evaluation in this study is conducted by an expert questionnaire to confirm the completeness of platform building and easy access through the interface.

The objects in the expert questionnaire interview were designed by developers with experience in platform management and experts or architects with experience in historic building restoration, and computers are provided to them for standardized testing, including the installation of the integration platform and external applications. The evaluation participants were informed of the purpose and overall architecture of the integration platform and plug-in interface application, asked to actually operate the platform and plug-in interface, and then, completed the questionnaire after the operation process.

### 5.3.1. Questionnaire Design

The questionnaire in this study was divided into four parts. Part I is the basic survey data to collect the identity, background, and use experience of digital auxiliary tools, and Parts II and III are the accessibility evaluations of the platform interface and plug-in application interface, respectively. The accessibility evaluation adopted in this study is adapted from the System Usability Scale (SUS), as proposed by Brooke [25], and is composed of 10 statements, where half of the problems are expressed in positive terms, while the other half are expressed in negative terms. The scores calculated in the SUS represent the user's comprehensive evaluation of the system's accessibility. The last part is related to the background of platform administrators and experts.

In terms of the usability evaluation of the platform interface, the interactive experience provided by the Unity integration platform and the simplicity of the operation interface are evaluated to understand (1) whether the integration platform can effectively interact with information models, (2) whether the integration platform can provide correct model restoration information, (3) whether the interface is easy and intuitive to use, (4) whether the interface switch and menu design are logical, (5) whether the threshold of operating

the BIM software can be effectively reduced, and (6) whether it can effectively assist the design process of historic building restoration.

In terms of the accessibility evaluation of the plug-in application interface, understand the functions provided by the iData+ application interface (1) whether it can rapidly export and update the information to improve the accuracy of information collection; (2) whether the connection between the iData+ application interface and Unity integration platform can realize instant information transmission; (3) whether the iData+ application program provides added value to information; (4) whether the iData+ application program provides a simple and logical operation process.

In the last part, interviews were conducted with the platform administrator, experts, or architects of historic building restorers to understand from the perspective of the platform administrator: (1) whether the platform can easily import historic information models and integrate the information and (2) whether the platform is free and feasible in development. From the perspective of experts, it was evaluated (1) whether the restoration information content and functions provided by the platform can effectively assist the needs of the historic restoration design process and (2) whether it is conducive to on-site communication and application.

### 5.3.2. Questionnaire Statistical Results

This section is explained based on the above expert interview results. First, the questionnaire results are divided into six grades according to the SUS calculation method [26], which presents the calculation and evaluation results of the usability of the system built in this study. According to the statistical results of the questionnaire (Figure 9), the Unity integration platform helps to expand the application of the BIM models of historic information, which uses game technology to visualize information and develop interactive functions, strengthens the interaction between the user and information, includes restoration information for added value, and effectively reduces the difficulty of traditional BIM operation. The dynamic and static perspective switch provided by the platform improved the interactive experience between the user and the model, and information can be accurately transmitted between Revit and the Unity integration platform through the iData+ application interface. The automatic bi-directional information transmission function makes the integration process more logical and easier to operate. The interface application design is relatively user-friendly and achieves the design purpose of providing the user with information interaction. Intensifying the application of value-added information brings more possibilities for the restoration and conservation of historic buildings in the future.

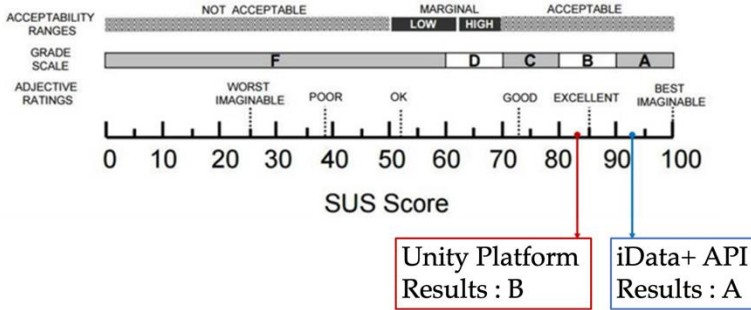

**Figure 9.** Statistical results of the questionnaire.

Regarding the platform management end, the workflow of importing the models and information into the integration platform may be simplified through the iData+ application interface, which also reduces the development complexity of application functions and programs. More auxiliary functions can be developed in the future, including classifying the restoration information and building the automatic information classification function according to the restoration craftsmen or the different periods of restoration. At present, users are required to complete input fields; however, if the workflow can be automated,

the application to historic buildings will be more diversified, and the feasibility of big data application will be improved.

In addition, regarding the option of the "visual information list", as provided by the Unity integration platform (Figure 7), the relationship between the physical model component and corresponding component name button is not intuitive enough; thus, users cannot directly click on the physical model, but must find the corresponding component name button in the information list before pressing the button, which is not convenient or intuitive for the user. In addition, as the interface switch between Unity and Revit platforms requires supplementary instructions, this system is not easy enough for beginners, but usage may be improved as the user becomes familiar with the operation process.

This research focuses on the combination of the Revit plug-in interface application and the Unity platform. It is an experimental interactive interface development. The current preliminary results are evaluated by expert questionnaires. In this questionnaire, architects with experience in restoration of historical sites and experts with experience in system development are selected as the subjects of the questionnaire. The limited number of architects with experience in restoration of historical sites does limit the number of questionnaires and also affects the validity. However, the evaluation recommendations obtained from the results of the current research phase can still be expected as a reference for future research and subsequent system improvements (see Section 6.2 for the reference).

## 6. Results and Future Studies

### 6.1. Results

This study discusses the feasibility of the application of the BIM models of historic buildings in restoration and completes two tasks: building the Unity integration platform and developing the iData+ plug-in application. As the interactive function of the BIM models of historic buildings in the Revit platform is expanded, the visual application of historic building restoration information is strengthened through the Unity integration platform, which provides an intuitive operating interface that lowers the barriers to using the Revit BIM model.

The iData+ plug-in application developed in this study can overlay and update restoration information, and the generated restoration process information can be exported in the Excel format, which can be edited; thus, the information can provide added value through word processing software. Meanwhile, the cloud service framework setting is also completed for bi-directional conversion and real-time updating between Revit and Unity platforms. Information transmission between Revit and Unity is connected by the cloud end service architecture to upload the latest restoration information and simplify the complexity of the workflow.

As historic building components are very complex, and the model and information are not easily read, the "visual information list" option in the Unity integration platform can be accessed at the historic building restoration site for information comparison with the actual components. In addition to presenting the information and model in an interactive visual manner, users can use the "interactive information list" option to change and increase the restoration suggestions offered in the restoration information of the components to promote the application of the restoration information. Moreover, the "interactive automation" option is set, component disassembly steps and descriptions are provided, and disassembly automation is animated to facilitate communication and discussion between the design end and the construction end of the component restoration, in order to ensure that all parties have a correct understanding of the model information, which will improve the overall flow and cooperation.

This study used Unity as the integration platform to develop BIM model functions of historic buildings. When applied in the restoration design of historic buildings, visual information integration helps improve the interactive experience and follow-up management and increases the feasibility of preserving cultural heritage. The following are the three results and contributions of this study:

1.  Provide a visual environment for interaction with models and information: through the development of the Unity integration platform, strengthen the possibility of future applications of historic building restoration designs and provide an interactive platform to assist the restoration design process and communication.

2.  Real-time automatic bi-directional information transmission: the iData+API plug-in application built in this study can upload revised information and carry out bi-directional overlaying and updating, in order to enrich historic building restoration information and simplify the transmission of manual information.

3.  Maintain the sustainable application of model information: through the information collection and programming integration process, the transmission and integration of models and information between different platforms can be achieved, and the information transmission of a cloud service framework can make it easier to import and apply the information model of historic buildings.

### 6.2. Future Studies

The Unity integration platform and iData+ plug-in application interface were developed in this study to develop the user's interaction with information, expand the visual application of the BIM models for historic buildings, assist in meeting demands for increased restoration design communication, and strengthen the link between complex component models and information. However, there are still problems remaining for the user, software, and technology in the solutions of this study, which are described as follows.

In terms of the user, this study mainly focused on assistance in the restoration design end: as the construction end is weak, if the platform functions can be developed in more directions and the information content needed in the construction process can be provided, the usability of the integration platform can be improved and the restoration process of historic buildings will be assisted in a more accurate and efficient manner.

In terms of software, since personal computers or laptops are used as tools for the operation platform, and the interface operation is also designed based on mouse use, in order to integrate the platform at the restoration site, a personal computer must be available, which can be relatively inconvenient.

In terms of technology, animation simulation for disassembly is currently presented in a visual and interactive manner. If the knowledge records of restoration craftsmen can be added to the interactive animation function, the restoration process of historic building models can be viewed with auxiliary visual knowledge tools. Moreover, when the components are disassembled, the disassembled components can be coded and the location of the operation area can be recorded to facilitate subsequent management and reassembly, which would improve management capability and restoration quality.

The design of the integration platform should not be limited to the operation of computers; thus, in the future, portable mobile phones, tablets, and other digital products can be added, to ensure that restoration designs can be communicated and discussed without time or location limits. In the future, more application functions can be developed to optimize the restoration process at the construction site, including automatic classification of restoration information, setting the information classification function for restoration at different stages, adding photos at the construction site for comparison and review, and intensifying the application of historic building information. In addition, the Unity integration platform developed in this study contains no geographic location information; thus, in the future, the author plans to integrate building information modeling (BIM) and a geographic information system (GIS) for large-scale integrated applications of big data information more suitable for historic blocks.

**Funding:** This research was funded by the Ministry of Science and Technology of Taiwan, grant number MOST 108-2621-M-390-001.

**Informed Consent Statement:** Not applicable.

**Data Availability Statement:** Not applicable.

**Acknowledgments:** The author is grateful for the generous grant by the Ministry of Science and Technology of Taiwan, coded number: MOST 108-2621-M-390-001. Yu-Shan Li is greatly appreciated for her technical assistance in the development of the system. The author would also like to express their deepest appreciation to all those who provided them with the possibility to complete this report.

**Conflicts of Interest:** The author declares no conflict of interest.

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
