# Peer review of "Improved Interaction of BIM Models for Historic Buildings with a Game Engine Platform"

_applsci, doi:10.3390/app12030945_

Round 1

Reviewer 1 Report

The topic discussed in this paper is highly relevant in several field of research connected to built heritage. In the presented case study, authrs analyse a wooden structure, also in the abstract authors make it explicit that this procedure has been applied to just one very specific case study, it would be interesting to understand if the same procedure would be applicable to a wider range of existing buildings and if there are specific limits related, for example, to the scale of the object. However, the paper is very well structured and has the merit to find a possible pathway for an easier and wider communication of cultural heritage. On the other end, for what concerns research and dissemination, specialists, such as restorers or historians, would probably still need to access specialistic models in which they can have a more detailed and flexible visualization of characteristics related to the state of conservation of the single object, the metrical reliability of the elements and  also, for example, survey documentation related to archaeological remains connected to the building. 

Author Response

Dear Sir

Thank you for your comments and suggestions, the following is my response:

Regarding the evaluation of the platform: This research focuses on the combination of the Revit plug-in interface application and the Unity platform. It is an experimental interactive interface development. The current preliminary results are evaluated by expert questionnaires. In this questionnaire, architects with experience in restoration of historical sites and experts with experience in system development are selected as the subjects of the questionnaire. The preliminary suggestion is to provide a reference for future research to be revised. At the same time, the questionnaire chapter is placed under the fifth chapter and presented in chapter 5.3. Please see Line 704-793 for the reference.

Reviewer 2 Report

Dear Autors, this is a good article, well written and with a clear and well articulated structure. It deals with innovative topics of keen interest in the scientific community such as the use of BIM and the overcoming of certain limitations by exploiting the potential offered by the world of game engines. 

The article describes the entire workflow, from the construction of the BIM to its implementation in the Unity software, as well as interface design and evaluation aspects. 

However, some information is missing in the text to make the workflow clearer and more transparent. Here are some points for further discussion:

1) It would be useful for the reader if the authors would disclose some information about the timing and method of development: How long does it take to export the model from Revit and import it into Unity?  What kind of 3D modelling issues have you encountered (normals, topologies, materials, etc.)? How do the virtual navigation controls work (especially in immersive VR environment)?

2) Regarding the evaluation of the platform: How many persons has it been tested on? What kind of users has the platform been tested on? It would be useful if the authors presented a small case study explaining the practical advantages obtained in the analysis aimed at restoration planning.

3) In general, the images are well done and schematise the work rather well, but they do not do illustrate completely the functioning of the platform. It would be useful to have some clearer screenshots or a video of the application in order to better illustrate the added value of the system indisputable compared to a more traditional use. The authors should provide, as additional material, a video demonstrating the functioning of the platform and in particular how: 
a)  it improves interactive communication
b) it allows to navigate, interrogate, dismantle buildings, etc.

4) Why is there no mention of H-BIM (heritage Building Information Modelling) in the text? Usually this term is used to refer to BIM of historical buildings.

Thank you for you contribution.

Author Response

Dear Sir

Thank you for your comments and suggestions, the following is my response:

Comments: 1) It would be useful for the reader if the authors would disclose some information about the timing and method of development: How long does it take to export the model from Revit and import it into Unity?  What kind of 3D modelling issues have you encountered (normals, topologies, materials, etc.)? How do the virtual navigation controls work (especially in immersive VR environment)?

The response to the above comment:

Please see Line 573-592 for the response to the above comment.

Comments: 2) Regarding the evaluation of the platform: How many persons has it been tested on? What kind of users has the platform been tested on? It would be useful if the authors presented a small case study explaining the practical advantages obtained in the analysis aimed at restoration planning.

The response to the above comment:

Please see Line 704-793 for the response to the above comment. Regarding the evaluation of use, this research focuses on the combination of the Revit plug-in interface application and the Unity platform. It is an experimental interactive interface development. The current preliminary results are evaluated by expert questionnaires. In this questionnaire, architects with experience in restoration of historical sites and experts with experience in system development are selected as the subjects of the questionnaire. Due to the limited number of architects with experience in restoration of historical sites, it does limit the number of questionnaires and also affect the validity. The preliminary suggestions can still be provided a reference for future research (see the chapter 6.2 for the reference) to be revised. At the same time, the questionnaire chapter is placed under the fifth chapter and presented in chapter 5.3.

Comments: 3) In general, the images are well done and schematise the work rather well, but they do not do illustrate completely the functioning of the platform. It would be useful to have some clearer screenshots or a video of the application in order to better illustrate the added value of the system indisputable compared to a more traditional use. The authors should provide, as additional material, a video demonstrating the functioning of the platform and in particular how: 
a)  it improves interactive communication
b) it allows to navigate, interrogate, dismantle buildings, etc.

The response to the above comment:

Please see Figure 7 & 8 for the response to the above comment. Fig. 7 and Fig. 8 are an example of a user executing an integrated platform project in the repair site.

Comments: 4) Why is there no mention of H-BIM (heritage Building Information Modelling) in the text? Usually this term is used to refer to BIM of historical buildings.

The response to the above comment:

This term H-BIM indicates a new way of modelling existing buildings, using a BIM process that would produce intelligent models containing and managing information. HBIM modelling starts from a survey: usually a digital survey that is achieved by using tools like a laser scanner or a camera for land photogrammetry. The so-called point clouds, sets of points with defined coordinates in space, are thus obtained. This research does not focus on modelling existing buildings, but on developing a 3D interactive information management platform that allows information query and virtual navigation. So the concept of H-BIM (heritage Building Information Modelling) is not specifically mentioned here.

Reviewer 3 Report

The paper presents a system that integrates BIM technology for historic buildings integrated into an interactive environment developed in Unity. The system describes the Unity-Revit integration and discusses the results and benefits of such an implementation.
First of all, I would like to commend the authors for their interesting and useful approach for Architects and engineers working on the reconstruction and maintenance of historical buildings.
However, the paper is not free of drawbacks, being its excessively verbose writing the most important one. First of all, let me clarify that I have found the English to be almost prefect and without noticeable mistakes. However, the sentences are extremely redundant, and the overall paper repeats several times the same concepts, over and over again. Basically, the paper should be trimmed to about half its current size before even being considered for publication. One example is the previous work section (Research Background), which is 5 pages long, 25% of the whole paper. This is too much, should be shortened quite a bit. By the way, the previous work should be much more than a simple enumeration of the existing literature, it should position this paper in the scientific context. Simply enumerating papers does not provide much information about the advantages and disadvantages of this paper with respect to the state-of-the-art. Another example is the last paragraph before Section 2.1, which almost does not provide any information, besides repeating some keywords several times (e.g., game technology platform, historic buildings, …). Other sentences, such as “Therefore, as the BIM model for historic buildings still lacks an environment that can interact with the model, new possibilities can only be provided to BIM by importing them into the development of other environments.”, are difficult to parse. Another example is the first two sentences in the second paragraph of Section 2.3, which use the word “therefore” to introduce something that is NOT a consequence of the previous one. Please, correct.
Another drawback, in my opinion, is the awkward structure the overall paper has. For instance, it directly starts with the previous work, without any kind of introduction of explanation of why any reader should care about this whole problem. Please, add a proper introduction explaining what the problem is, and giving the main contributions upfront, so the interested reader can know before reading the whole article what is the paper objective and main results.
Next, the paper discusses a user study, but it does not present more information that Figure 9. Please, indicate number of users, average expertise, demographics, ages, etc. Also, what is the confidence interval of these results? Please, provide a solid, well-funded statistical analysis of these results. As presented now, it seems an arbitrary analysis without much ground.
Finally, as the previous work is written now, I cannot appreciate the differences with the previous work from the same author. Please, clarify what are the differences and similarities between this work and the previous ones by the same author.
Other minor comments:
* Please, insert the references just after the name of the authors has been introduced, for instance Ma, Y.P. et al [reference], which eases reading and looking for a given paper.
* In Figure 2, what does the numbers 1 and 2 in circles in the image mean?
* I do not understand the usefulness of the Web App. What is this for? It seems out of context…
* The first sentences of Section 4.1 are clearly repeated from previous sections of the paper. Please, rephrase.
* Please, clarify the use and utility of the iData+ description, because, as it is now, it seems to make all previous descriptions a bit useless. Please, justify.

Author Response

Dear Sir

Thank you for your comments and suggestions, the following is my response:

1. Regarding the issue of the awkward structure the overall paper:

Please see Line24-66 for the response to the above comment. I have revised the entire chapter structure, moved the content originally placed between chapter 2 and 2.1 to the beginning of the article for proper introduction, raised questions, and gave the main contributions in advance. Help readers know the purpose and main results of the paper before reading the entire article.

2. Regarding the previous work should be much more than a simple enumeration of the existing literature, it should position this paper in the scientific context:

Please see Line67-289 for the response to the above comment. The previous works are moved to chapter 2 literature review and are discussed in three subsections. Each subsection describes the phased research results of the introduction of different digital technologies, and their advantages and disadvantages, to help readers better understand the research context of this research.

3. Regarding the evaluation of the platform:

Please see Line704-793 for the response to the above comment.

This research focuses on the combination of the Revit plug-in interface application and the Unity platform. It is an experimental interactive interface development. The current preliminary results are evaluated by expert questionnaires. In this questionnaire, architects with experience in restoration of historical sites and experts with experience in system development are selected as the subjects of the questionnaire. The preliminary suggestion is to provide a reference for future research (see the chapter 6.2 for the reference) to be revised. At the same time, the questionnaire chapter is placed under the fifth chapter and presented in chapter 5.3.

4. Regarding the differences with the previous work from the same author.

The two previous works mentioned in the article [5] & [20], the former completed the development of the Revit plug-in interface API, and the latter began to try to import the information model into the Unity platform for the integration of geometric models and non-geometric semantics. This research uses the former's Revit information model [5], based on the former's experience in developing the Revit API and the latter's experience in importing the model into the Unity platform [20], and tries to combine the Revit API with the Unity platform and develop an interactive interface. Please see Line137-147 & 258-269 for reference.

5. Regarding other minor comments:

(1)Please, insert the references just after the name of the authors has been introduced, for instance Ma, Y.P. et al [reference], which eases reading and looking for a given paper.

Please see Line 192, 201,220,241,248,261 for the response to the above comment.

(2) In Figure 2, what does the numbers 1 and 2 in circles in the image mean?

Please see Line 505,507,515 for the response to the above comment. The symbols are added to the text to explain the numbers 1 and 2 in circle for the convenience of readers to compare.

(3) Description about the use and utility of iData+ api &Webapi.

Please see Line 543-552, 655-664, 655-664, 696-703 for the response to the above comment.

Round 2

Reviewer 3 Report

  I've checked the revised version, and I am afraid it only PARTIALLY covers my concerns. First of all, the answer about the differences with respect to the author's own work should be on the paper itself, not only on the response letter. Second, the author simply ignored my request for more accurate characterization of the user study. I think that, without a proper description of such a study, we cannot proceed with the publication of the article. Finally, I still think the paper is overly too verbose and should be significantly shortened, but if the editorial team accepts such a lengthy paper, I would not complain.

Author Response

Dear Sir

Regarding your concerns, my response is as follows:

  1. Regarding our previous works, please see Line 290-298 for the response.
  2. Regarding the evaluation of use, please see Line 797-804 for the response.
  3. Regarding the paper is overly too verbose, please see Line 67-298 for the response.

Thank you again for all your suggestions. Wishing you a very joyful New Year 2022. :-)
